# SNP Array Screening and Long Range PCR-Based Targeted Next Generation Sequencing for Autosomal Recessive Disease with Consanguinity: Insight from a Case of Xeroderma Pigmentosum Group C

**DOI:** 10.3390/genes14112079

**Published:** 2023-11-15

**Authors:** Fumie Nomura, Akira Shimizu, Sumihito Togi, Hiroki Ura, Yo Niida

**Affiliations:** 1Department of Dermatology, Kanazawa Medical University, Uchinada 920-0293, Japanashimizu@kanazawa-med.ac.jp (A.S.); 2Center for Clinical Genomics, Kanazawa Medical University Hospital, Uchinada 920-0293, Japanh-ura@kanazawa-med.ac.jp (H.U.); 3Department of Advanced Medicine, Division of Genomic Medicine, Medical Research Institute, Kanazawa Medical University, Uchinada 920-0293, Japan

**Keywords:** SNP array, isodisomy, autosomal recessive, consanguinity, long range PCR-based NGS, xeroderma pigmentosum group C

## Abstract

Advances in genetic technologies have made genetic testing more accessible than ever before. However, depending on national, regional, legal, and health insurance circumstances, testing procedures may still need to be streamlined in real-world clinical practice. In cases of autosomal recessive disease with consanguinity, the mutation locus is necessarily isodisomy because both alleles originate from a common ancestral chromosome. Based on this premise, we implemented integrated genetic diagnostic methods using SNP array screening and long range PCR-based targeted NGS in a Japanese patient with xeroderma pigmentosum (XP) under the limitation of the national health insurance system. SNP array results showed isodisomy only in *XPC* and *ERCC4* loci. NGS, with a minimal set of long-range PCR primers, detected a homozygous frameshift mutation in *XPC*; NM_004628.5:c.218_219insT p.(Lys73AsnfsTer9), confirmed by Sanger sequencing, leading to a rapid diagnosis of XP group C. This shortcut strategy is applicable to all autosomal recessive diseases caused by consanguineous marriages, especially in scenarios with a moderate number of genes to test, a common occurrence in clinical genetic practice.

## 1. Introduction

Genetic testing is a cornerstone of clinical genetics, providing accurate diagnoses, enabling precision medicine, and providing essential information for genetic counseling. Targeted sequencing is performed when the clinical diagnosis is clear and the number of genes to be analyzed is limited to a small number. Comprehensive tests, such as gene panels and whole exome sequencing (WES), are used when the number of genes to be analyzed is large or when the clinical diagnosis is difficult to confirm. There is also a growing use of whole genome sequencing (WGS) [1]. In an intermediate situation, where the clinical diagnosis is confirmed but there is a moderately large number of genes to be analyzed (10 to 20) and no appropriate gene panel, it can be confusing to choose an analysis method. There is a strategy to perform WES and analyze only the necessary genes and use it as a virtual panel [2], but this is not always possible due to differences in countries and regions, laws, and health insurance. First of all, with this method, most of the analyzed data are wasted and remain unused. In actual clinical settings, it is sometimes necessary to streamline genetic testing, even today.

In cases of autosomal recessive diseases within a consanguineous family, the inheritance of both homozygous mutation alleles from a single ancestral chromosome, coupled with one or two crossing over per chromosome per successive generations [3,4], leads to segmental isodisomy around the mutated gene (Figure 1). Therefore, single nucleotide polymorphism (SNP) microarrays can effectively screen the locus. However, once the locus is identified, the challenge shifts to testing the candidate gene. Whether designing PCR primers for individual exons and performing direct sequencing or designing capture probes for specific genes and using targeted sequencing on next-generation sequencers (NGS), some effort and expense are required. A viable shortcut is to use targeted NGS with long-range PCR. This approach allows a comprehensive study of the entire genomic region with a minimal number of PCR primer sets, and by utilizing Nextera technology (Illumina, San Diego, CA, USA), libraries can be prepared directly from the long PCR products [5,6].

Xeroderma pigmentosum (XP) is an autosomal recessive disorder characterized by an acute sensitivity to sunlight, a significantly increased risk of cutaneous neoplasia, and ocular manifestations (photophobia, severe keratitis, eyelid skin atrophy), and approximately 25% of affected individuals have neurological manifestations, including hearing loss and progressive motor and cognitive impairment [7,8,9]. Nine causative genes (*DDB2*, *ERCC1*, *ERCC2*, *ERCC3*, *ERCC4*, *ERCC5*, *POLH*, *XPA*, *XPC*) are known for XP. Several other genes causing autosomal recessive nucleotide excision repair disorders exhibiting cutaneous photosensitivity (*ERCC6*, *ERCC8*, *GTF2H5*, *GTF2E2*, *MPLKIP*, *UVSSA*) are also included in the differential diagnosis [7]. Since each of these genes has different neurological implications, precise differentiation between them is crucial. Here, we present a patient with xeroderma pigmentosum who was molecularly diagnosed as group C using a rapid strategy of initial screening by SNP array, followed by confirmation of an XPC mutation by long range PCR-based targeted NGS.

## 2. Materials and Methods

### 2.1. Case Presentation

This case involves a 65-year-old woman whose parents were cousins. Since childhood, she has had a mixture of pigmented and depigmented macules on her trunk and extremities. She has a history of treatment for psoriasis vulgaris. About 1 year ago, she noticed a nodular lesion on her left upper eyelid and consulted the dermatology department of our hospital. A brown to blackish flat nodule was observed on the medial side of the left upper eyelid (Figure 2a). Histopathologic findings revealed that basophilic tumor cells had formed a cyst and were proliferating within the dermis. Peripheral palisading and clefting were also observed (Figure 2b). The patient was diagnosed with basal cell carcinoma. Despite the absence of ocular or neurological symptoms and no hypersensitivity symptoms on light testing, XP was suspected because of the presence of multiple pigmented macules on the face, trunk, and extremities (Figure 2a), the history of basal cell carcinoma, and consanguinity. To confirm the diagnosis, genetic testing was performed after genetic counseling at the Center for Clinical Genomics in our hospital.

### 2.2. Methods of Molecular Diagnosis

#### 2.2.1. Mutation Analysis

Genomic DNA was extracted from the patient’s peripheral blood using a rapid extraction method [10]. This method is capable of extracting very high molecular weight DNA and is suitable for long-range PCR amplification. The amount of DNA and the ratio of optical density (OD) A260/280 were measured using a Nanodrop instrument (Thermo Fisher Scientific, Waltham, MA, USA).

SNP array analysis was performed using a CytoScan 750K array (Thermo Fisher Scientific, MA, USA) and analyzed using Chromosome Analysis Suite (ChAS) 2.1 Software (Thermo Fisher Scientific). Regions larger than 1Mb and containing 100 or more SNP probes were extracted as loss of heterozygosity (LOH) segments.

The long-range PCR primers used in this study were designed by Primer3 v.0.4.0 (https://bioinfo.ut.ee/primer3-0.4.0/, last accessed 20 October 2023) [11], using the following parameters: primer length, 26–27–30 mer; Tm, 67 °C–67.5 °C–68 °C; Max Tm difference, 0.1 °C; and GC%, 45–50–60. GC Clump 2 and other parameters were used with default settings. Long PCR primer sets were designed to overlap with a length of approximately 20 kb to cover the entire genomic region of the target genes. The PCR primer sets constructed in this study are shown in Appendix A. Each PCR reaction contained 1 µL of 20 ng/µL genomic DNA, PCR primers at a final concentration of 0.15 µM, and KOD One DNA polymerase (TOYOBO, Osaka, Japan) in a 10 µL reaction volume. Touch-down PCR cycles were performed under the following conditions: 3 cycles of 98 °C for 10 s and 74 °C for 10 min; 3 cycles of 98 °C for 10 s and 72 °C for 10 min; 3 cycles of 98 °C for 10 s and 70 °C for 10 min; and 25 cycles of 98 °C for 10 s and 68 °C for 10 min, with a total run time of 5 h 58 min.

Purification and size selection (>1000 bp) of long PCR products was performed using an AMPure XP (Beckman Coulter Life Sciences, San Jose, CA, USA), with 0.4 × volume of AMPure XP mixed with the PCR products. An NGS library was prepared using an Illumina DNA prep with enrichment kit (Illumina, San Diego, CA, USA), according to the manufacturer’s protocol. Libraries were quantified using an HS Qubit dsDNA assay (Thermo Fisher Scientific) and a TapeStation 4200. Qualified size distributions were checked on a TapeStation 4200 using High Sensitivity D1000 ScreenTape. A 12.5 pM library was sequenced on an Illumina MiSeq system (2 × 250 cycles) using the MiSeq Reagent Nano Kit v2 (500 cycles), according to the standard Illumina protocol (Illumina).

The FASTQ files were generated using the bcl2fastq software (Version 1.8.4) (Illumina). The FASTQ files were aligned to the reference human genome (hg38) using the Burrows–Wheeler Aligner MEM algorithm (BWA–MEM version 0.7.17-r1188) [12]. Haplotype variant calling for a single sample was performed using GATK’s HaplotypeCaller (Version 4.0.6.0) [13]. The SNVs and INDELs were functionally annotated by SnpEff (Version 4.3t), to classify each variant into a functional class (HIGH, MODERATE, LOW, and MODIFIER) [14]. The Database of Short Genetic Variations dbSNP (Version 151) and ClinVar were used for variant annotation [15,16]. For visualization, the Integrative Genomic Viewer (IGV version 2.4.13) was used [17].

The detected mutation was validated by direct DNA sequencing using the *XPC* exon 2 specific primer set (Appendix A) and BigDye Terminator v3.1 cycle sequencing kit with the ABI PRISM 3100xl Genetic Analyzer (Thermo Fisher Scientific).

#### 2.2.2. Immunohistochemical Staining of XPC Protein

After the identification of the XPC frameshift mutation, immunohistochemical staining was performed on formalin-fixed paraffin-embedded (FFPE) skin tissue samples obtained from the patient and healthy controls using a polymer peroxidase kit (Nichirei, Tokyo, Japan). Briefly, tissue sections were autoclaved in ethylenediaminetetraacetic acid (EDTA) at pH 9.0 for 15 min, followed by a cooling period of 30 min. The sections were then incubated overnight with a mouse IgG2a XPC monoclonal antibody (XPC (D-10) sc-74410, raised against the C-terminal amino acids 641–940 of human XPC, obtained from Santa Cruz, CA, USA).

## 3. Results

### 3.1. SNP Array

In the SNP array, the chromosomal region of isodisomy is represented as copy-neutral LOH. Hence, the SNP array measures both copy number and SNP combination type (AA, AB, or BB) across entire chromosomes. Isodisomy regions are shown as two-copy regions without hetero SNPs (AB) (purple boxes in Figure 3). The genes causative or related to XP were mapped on the result of the SNP Array. Only *XPC* and *ERCC4* were located in isodisomy regions.

### 3.2. Long Range PCR-Based NGS and Validation of Sanger Sequencing

Based on the SNP array results, long range PCR-based targeted NGS was performed for *XPC* and *ERCC4*. Entire genomic lesions of both genes were well amplified, and a homozygous frameshift mutation was detected in *XPC*: NM_004628.5:c.218_219insT p.(Lys73AsnfsTer9) (Figure 4a). This mutation was validated by Sanger sequencing (Figure 4b). No pathogenic mutation was detected in *ERCC4*, and all detected polymorphisms in these genes were homozygous and concordant with isodisomy (Appendix A).

### 3.3. Immunohistochemistry

To clarify the expression of XPC protein, immunohistochemical analysis using an anti-XPC antibody was performed. The results revealed significantly weaker XPC staining in the skin of the patient compared with the control, indicating a diminished expression of the XPC protein attributable to the frameshift mutation (Figure 2c).

## 4. Discussion

XP is an autosomal recessive disease caused by mutation of nucleotide excision repair-related genes, characterized by acute sun sensitivity and a significantly increased risk of cutaneous neoplasia, ocular manifestations, and neurologic manifestations [7,8,9]. The condition exhibits significant genetic heterogeneity, traditionally classified through hybridoma complementation experiments. Presently, nine genes are associated with XP, along with several others requiring differential diagnosis. Genetic diagnosis is crucial due to variations in the severity and frequency of complications, particularly neurological symptoms, depending on the causative gene.

In the reported patient, the disease-causing mutation was detected in *XPC*; it was confirmed that the patient would not suffer from neurological symptoms throughout her life. The detected pathogenic variant, NM_004628.5:c.218_219insT p.(Lys73AsnfsTer9), is located in exon 2 of the 16 exons of mRNA variant 1, and at the protein level, a frameshift occurs at the 73rd lysine among 940 amino acid residues, followed by the appearance of a stop codon at the 9th codon. Although analysis at the mRNA level has not been performed in this case, it is likely that the mutated mRNA is removed by nonsense-mediated mRNA decay (NMD) and no protein is produced [18]. The protein encoded by *XPC* is a key component of the XPC complex, which plays an important role in the early steps of global genome nucleotide excision repair (NER). The XPC complex consists of XPC, RAD23B, and CETN2, and the binding domains of the XPC protein with these partners are located at amino acid residues 496–734 and 847–863, respectively. Furthermore, the DNA-binding domain of XPC protein is present at amino acid residues 607–742 [19]. Therefore, even if this patient’s mutant XPC protein escaped NMD and was expressed, it would not be able to form an XPC complex or bind to DNA, and it would not exert any function. For immunohistochemical staining, an antibody that recognizes amino acid residues 641–940 on the C-terminal side was used, and as expected from the variant information at the DNA level, a significant decrease in staining was observed (Figure 2c).

As mentioned above, the number of genes causing XP or required for differential diagnosis is moderately large, which makes it difficult to select an analysis method when providing a genetic diagnosis as a clinical test under the social issue, for example, with the limitation of national health insurance coverage. Traditionally, genetic testing has focused on the causative gene of the disease based on clinical diagnosis. Genetic diagnosis is streamlined when the clinical diagnosis is clear from characteristic symptoms and the nature of the pathogenic mutations is clearly established. The most obvious example is achondroplasia, where nearly all patients have a common missense mutation, p.Cly380Arg, in the *FGFR3* gene. Therefore, single-point Sanger sequencing is sufficient for accurate genetic testing [20,21]. In situations of confirmed clinical diagnosis, where the number of causative genes is limited but there are no mutation hotspots, the traditional approach has been to perform Sanger sequencing of each coding exon as the gold standard. However, the efficiency of this sequential sequencing strategy diminishes when dealing with large target genes with numerous exons. A contemporary alternative is targeted sequencing using next-generation sequencing (NGS), as exemplified by its application in the cases of *BRCA1* and *BRCA2* for hereditary breast and ovarian cancer syndrome (HBOC) [22] or *NF1* for neurofibromatosis type 1 [23]. Gene panels are useful when dealing with diseases caused by a large number of genes due to high genetic heterogeneity, or when screening for genetic diseases based on organ-specific or disease category criteria. By expanding the target genes to include all genes, whole exome sequencing (WES) has emerged as a powerful diagnostic tool, particularly adept at elucidating undiagnosed diseases. This expanding research trajectory has now led to the era of whole genome sequencing (WGS) [1,24]. On the other hand, in clinical practice, a significant proportion of patients require testing of 10 to 20 genes, a relatively modest number, involved in their disease, making the selection of an appropriate testing method a challenge. In the absence of appropriate gene panels, the construction of a new panel is a laborious and costly task. Conversely, the choice of whole exome sequencing (WES) presents a dilemma because most of the sequencing data are not needed for diagnosis, resulting in a waste of resources. However, as the cost of WES has decreased, the use of WES as a virtual panel has increased [2]. This method has the advantage of reducing the effort required for variant interpretation by limiting the target genes to be analyzed, as well as the ability to freely change the genes included in the panel. As a result, the widespread use of gene panels and whole exome sequencing (WES) has greatly expanded the scope of genetic testing.

However, in real-world clinical practice, regulatory, insurance, and cost challenges continue to impede the seamless integration of advanced genetic testing technologies. Despite the diagnostic potential of whole exome sequencing (WES), barriers, particularly in the area of insurance coverage, pose significant barriers to access for a significant patient population. As noted by Reuter et al. [25], a survey within the Undiagnosed Disease Network found that 66 individuals (40%) out of 165 patients faced barriers related to insurance coverage that affected their ability to undergo WES. In addition, the imposition of test cost caps by national health insurance plans may impede the adoption of financially demanding methods such as WES. This underscores the need to make genetic testing more cost effective and streamlined within the practical context of clinical genetics, depending on actual national and regional circumstances.

In this case report, we present a patient with xeroderma pigmentosum who underwent genetic testing under the Japanese national health insurance system. In Japan, genetic testing for XP has been covered by national health insurance since 2022 but no disease-specific NGS panel is available, and the coverage cost for the test is approximately US $350. WES is being used in the Initiative on Rare and Undiagnosed Disease (IRUD) Japan study [26], but only for the patients whose diagnoses cannot be determined clinically. In addition, the medical law strictly distinguishes between research tests and clinical tests, and research tests cannot be used for clinical diagnosis in Japan. Furthermore, it is not possible to use national health insurance and personal health insurance at the same time during the same treatment period. Under these restrictions, it was necessary to streamline the methods used to perform genetic testing for this patient.

As shown in this report, gene loci responsible for autosomal recessive diseases associated with consanguinity can be screened using SNP arrays. In the early 2000s, SNP arrays played a pivotal role in numerous genome-wide association studies (GWAS), facilitating the identification of large numbers of disease-associated SNPs [27]. In recent years, the focus of GWAS research has shifted to the discovery of rarer disease-associated SNPs through the application of NGS [28]. Also, in the field of autosomal recessive diseases, SNP arrays have proven valuable in cases where the causative gene is unknown, applying homozygosity mapping to determine the disease locus [29,30]. SNP arrays have gained popularity as a pediatric genetic test because they offer a distinct advantage over comparative genomic hybridization (CGH) arrays. Unlike CGH arrays, SNP arrays can not only measure the excess or deficiency of chromosomes but can also assess isodisomy status. This capability has been used to classify the developmental mechanism of imprinted diseases, such as Prader-Willi syndrome, and has shown that some cases are caused by isodisomy of the chromosome from one parent in which the imprinted gene is present [31,32]. It has also been used to clarify autosomal recessive diseases where only one parent is a carrier and the disease is a result of the uniparental isodisomy [33]. In addition, SNP arrays serve as valuable tools to narrow down candidate genes for autosomal recessive diseases within consanguineous marriages, as exemplified in the presented case. Following the same concept, SNP array has been applied as the first step of molecular diagnosis in the highly heterogeneous muscle disease, autosomal recessive limb-girdle muscular dystrophy (LGMD2), in a consanguineous family [34] and autosomal recessive Charcot-Marie-Tooth disease in patients from inbred families [35]. They called this screening method “SNP-array based whole genome homozygosity mapping”.

After identifying candidate genes by SNP array, a target sequence for that gene is required to confirm the mutation. In recent years, capture sequencing using NGS has become mainstream for this purpose, but when analyzing new genes, it is necessary to set up capture probes, which requires a certain amount of cost and effort. Long range PCR-based NGS is one of the shortcuts to solving this problem and offers several advantages over the capture sequencing method. First, only a minimal number of long PCR primer sets need to be designed, enabling on-demand genetic testing. In fact, when XPC and ERCC4 were sequenced in this report, it was possible to completely cover each gene region with only two sets of long PCR primers (Appendix A). Second, compared with the capture probe method, the library preparation is faster, simpler, and devoid of off-target sequences, ensuring comprehensive coverage of all target regions. In addition, long range PCR-based NGS allows direct detection of breakpoint sequences for structural abnormalities such as large intragenic deletions [5]. The possibility that XP is caused by a large intragenic deletion cannot be excluded. According to a Gene Review, the proportion of large deletions or insertions in *POLH* pathogenic variants is estimated to be about 15% [7]. In Tunisia, several families have been reported with the XP variant caused by a 3925 base pair intragenic deletion, NG 009252.1: g.36847 40771del, including exon 10 of the *POLH* gene [36]. There are no reported cases of large intragenic deletions in *XPC* based on database searches. However, in Gene Review, the proportion of large deletion insertions in pathogenic mutations is unknown and cannot be excluded [7]. Therefore, when performing genetic testing for *XPC* and *ERCC4*, there is some value in screening the entire gene region, including introns, by long range PCR-based NGS. In addition to its advantages, long range PCR-based NGS is cost-effective, with a low per-sample analysis cost. Sequencing 25 samples simultaneously with 100 kb of target DNA per sample using the MiSeq Reagent NanoKit v2 (500 cycles) results in an average depth of more than 200, and the cost per sample is less than US $25, including DNA extraction and library preparation. In addition, when testing different genes in different patients, it is possible to combine PCR products and prepare a library using the same index, further reducing costs. Since the median size of a human gene on the genome is about 26 Kb [37], for many genes, one or two primer sets can cover the entire gene, including the promoter region, making it easy to set up new gene analyses on demand. In addition, when the size of the target gene is large, the procedure can be simplified by multiplex long-range PCR [5,6]. In our laboratory, where genetic testing for specific genes based on clinical diagnosis is common, long range PCR-based NGS has proven to be a valuable method, with long PCR primers already set up for over 250 genes.

Advances in genetic analysis technology have made genetic testing more accessible than ever before. However, not all national, regional, legal, and health insurance issues have been resolved. Even today, there are times when the efficiency of genetic testing needs to be streamlined. As shown in this case report, when there are many genes to be tested for autosomal recessive diseases caused by consanguineous marriages, theoretical SNP array screening followed by long range PCR-based NGS serves as an effective option to solve the problem.

## Figures and Tables

**Figure 1 genes-14-02079-f001:**
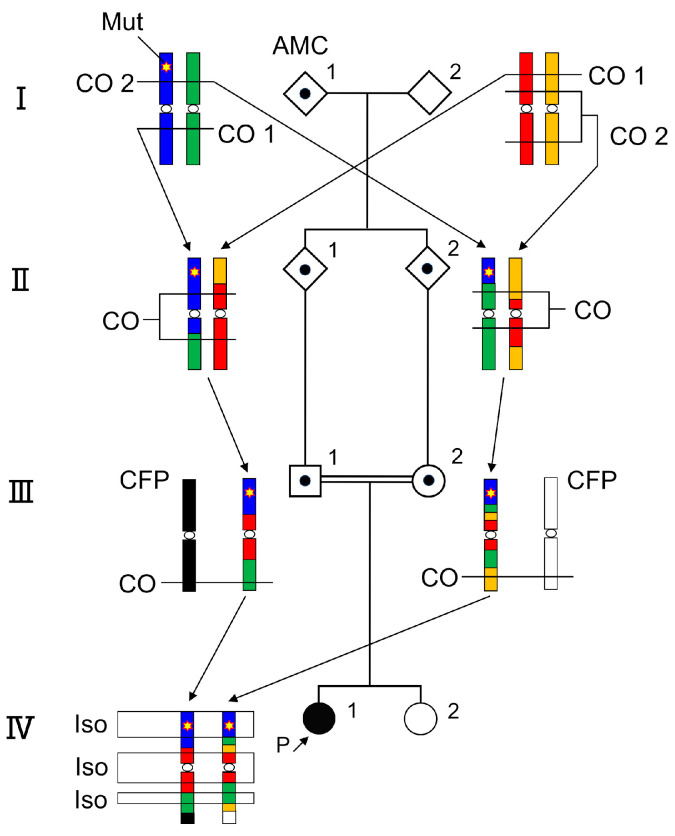
A homozygous mutation of an autosomal recessive disease is located at an isodisomy region in a consanguineous pedigree. The common mutation allele of the ancestral carrier (generation I) is inherited by siblings (generation II), cousins (generation III), and the patient (generation IV) with sequential chromosomal crossover. AMC: ancestral mutation carrier, CO: crossing over, CFP: chromosome from the partner, Is:; isodisomy, Mut: disease causative mutation.

**Figure 2 genes-14-02079-f002:**
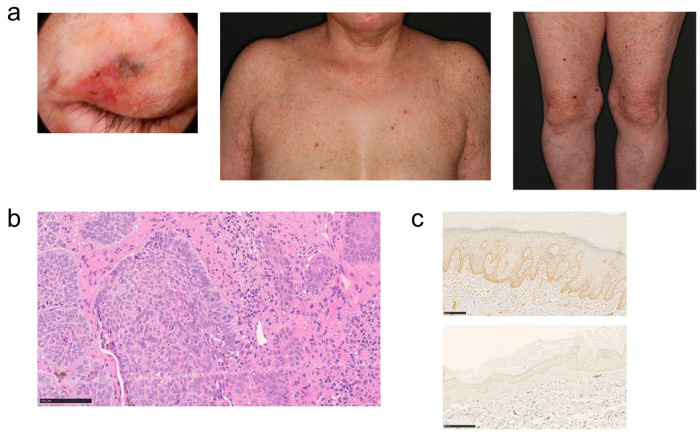
Skin lesions of the patient; (**a**) left eyelid, trunk, and lower extremities. (**b**) Histopathological image of left eyelid lesion with hematoxylin and eosin staining. (**c**) Immunohistochemical staining of XPC protein of skin tissue in controls (**top**) and patients (**bottom**).

**Figure 3 genes-14-02079-f003:**
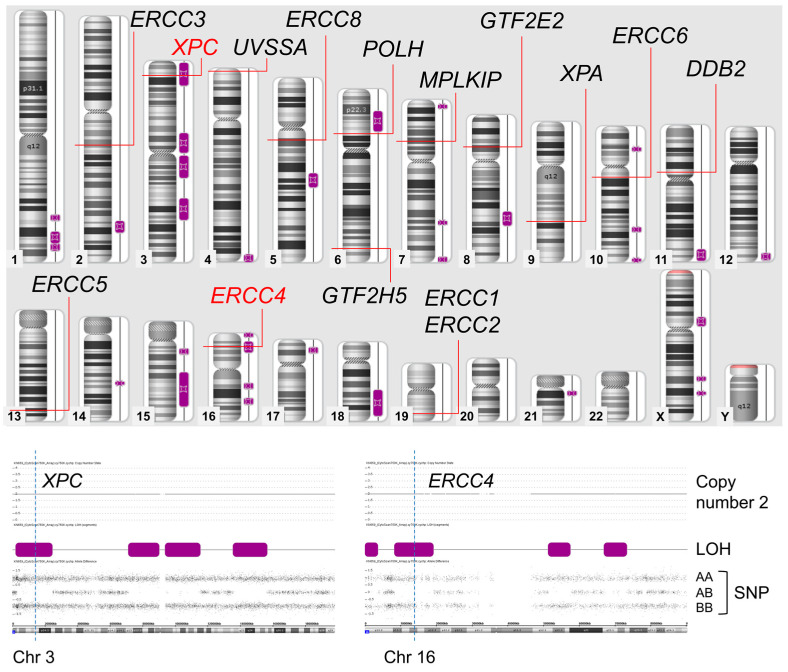
Results of SNP array. Only *XPC* and *ERCC4* genes are located on isodisomy regions. LOH; loss of heterogenicity, SNP; single nucleotide polymorphism.

**Figure 4 genes-14-02079-f004:**
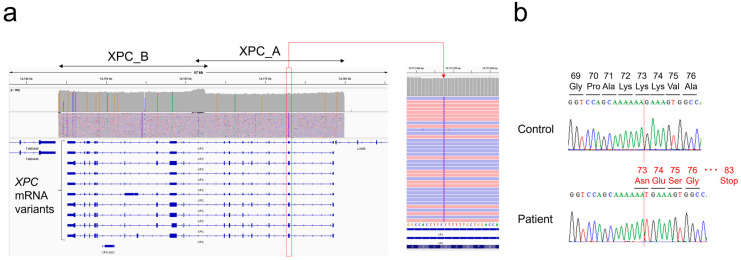
Determination of XPC frameshift mutation: (**a**) IGV image of long range PCR-based NGS. Note that the orientation of XPC is in the opposite direction on the genome. (**b**) Validation of the NGS result by Sanger sequencing.

## Data Availability

*XPC* pathogenic mutation reported in this study was registered in ClinVar (https://www.ncbi.nlm.nih.gov/clinvar/, last accessed 10 October 2023), with the submission ID SCV004037450.

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
