# Peer review of "SNP Array Screening and Long Range PCR-Based Targeted Next Generation Sequencing for Autosomal Recessive Disease with Consanguinity: Insight from a Case of Xeroderma Pigmentosum Group C"

_genes, 2023, doi:10.3390/genes14112079_

Round 1
Reviewer 1 Report
Comments and Suggestions for Authors
Introduction:
Line:33: the sentence: sequencing all individual genes is cumbersome is inappropriate. The best strategy for studing genetic etiology is exome sequencing o multigenepanel.
Line 40-42: Whether designing PCR primers for individual exons and conducting direct 40 sequencing or creating capture probes for specific genes and utilizing targeted sequencing 41 through next-generation sequencers (NGS), significant effort is required. The best strategy for studing genetic etiology is exome sequencing o multigenepanel.
See recommendations in the genereviews (https://www.ncbi.nlm.nih.gov/books/NBK1397/:
A xeroderma pigmentosum multigene panel that includes all of the genes listed in Table 1 and other genes of interest (see Differential Diagnosis) or exome sequencing
Results
3.2 3.2. Long-ranged PCR based NGS and validation of Sangar sequencing Sanger nor Sangar
Why authors did SNP arrays, the patient carried a homozygous fs variant in XPC. The first test should have been NGS and not an SNP array? The diagnosis is done using sequencing
It is not necessary fig1 and table 2 (polymorphisms). Table 1 should be inluded as supplementary data
Discussion
Lines 158-163: this paragraph about the use of SNP array for detecting uniparental disomy in Prader-Willi syndrome is not recommended. The EMQN guidelines recommend the use of microsatellites for chromosome 15. SNP array not detected heterodisomy (most frequent in PWS) unless parent’s samples are included, being a cost/effective strategy expensive.
In cases of families with consanguinity a strategy should be trio exome, especially when a recessive disease is suspected.
Lines 175-177
the variant detected is a frameshift no a deletion. A fs variant is caused by a deletion or insertion in a DNA sequence (in this case is a insetion) and the consequence is a premature termination of translation.
Is the variant described in scientific literature?
No correlation with phenotype?
Reviewer 2 Report
Comments and Suggestions for Authors
Minor observations:
sAbstract line 17:
“ the mutation locus must be isodisomy” modify these words, does not sound clear.
Lines 55-56 check and re-write the sentence
Lines 73-75 check and write better
Table 1 : consider erasing it from the paper and just write “primers available on request”
Lines 161-164: check both the English and if those reported are in fact the only diseases in which SNP arrays narrowing were used .
Line 175: “in our…” check language
Line 176-177: if this is in fact the ONLY additional XP-related gene in whom deletions have been reported, it should be stated clearly.
Line 179: “theolitically,” check language.
Major
The human gene XPC codes for a large protein (940aa; see UCSC database); the mutation found in the patient fall at Lys73AsnfsTer9; so only a very small protein, in theory , can be produced.
the picture (Fig2,c) seems convincing, but more details about the antibodies used (what in fact they recognize) should be provided;
similarly, how such a very small part of the protein (less than 10% of the normal protein) can in fact go trough all the steps of protein production till reaching the appropriate place in the cell, should be discussed extensively..
Comments on the Quality of English LanguageThe quality of English language should be improved; some sentences need to be red twice to be fully understood; please check minor comments to Authors.
Reviewer 3 Report
Comments and Suggestions for Authors
In this paper, Nomura et al., presented a patient with xeroderma pigmentosum, molecularly diagnosed as group C, through an initial screening via SNP array followed by confirmation of an XPC mutation through Long-range PCR-based targeted NGS.
Although it is an interesting case report, I'm not really agree about the exome sequencing is excessively wasteful. The authors could only have carried out the search on the proband with a selected panel of genes involved in xeroderma pigmentosum and could not perform SNP Array in all family members and a NGS long PCR.
Minor revisions:
Introduction: can the authors report about the similar cases described in the literature?
Figure 3: I suggest to do this figure again, It is a little blurry.
Table with oligos: I suggest to put them in Supplental files
Round 2
Reviewer 1 Report
Comments and Suggestions for Authors
No comments